# LANGUAGE MODELS CAN SELF-LENGTHEN TO GENERATE LONG TEXTS

## ABSTRACT

Recent advancements in Large Language Models (LLMs) have significantly enhanced their ability to process long contexts, yet a notable gap remains in generating long, aligned outputs. This limitation stems from a training gap where pre-training lacks effective instructions for long-text generation, and post-training data primarily consists of short query-response pairs. Current approaches, such as instruction backtranslation and behavior imitation, face challenges including data quality, copyright issues, and constraints on proprietary model usage. In this paper, we introduce an innovative iterative training framework called Self-Lengthen that leverages only the intrinsic knowledge and skills of LLMs without the need for auxiliary data or proprietary models. The framework consists of two roles: the Generator and the Extender. The Generator produces the initial response, which is then split and expanded by the Extender. This process results in a new, longer response, which is used to train both the Generator and the Extender iteratively. Through this process, the models are progressively trained to handle increasingly longer responses. Experiments on benchmarks and human evaluations show that Self-Lengthen outperforms existing methods in long-text generation, when applied to top open-source LLMs such as Qwen2 and LLaMA3.

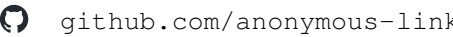 github.com/anonymous-link

## 1 INTRODUCTION

Recently we have witnessed significant breakthroughs in Large Language Models (LLMs), accompanied by extensive research aimed at improving their alignment with human demands (Cao et al., 2024; Gao et al., 2024; Quan, 2024a;b). One key research area is enhancing LLMs' abilities to manage increasingly long contexts (Han et al., 2023; Pawar et al., 2024; Bai et al., 2023). While much of the existing research on long context LLM alignment emphasizes processing lengthy inputs, and several open-source models like Qwen2 (Yang et al., 2024) and Llama3 (Dubey et al., 2024) herds of models already demonstrate impressive capabilities in long context understanding tasks, such as Needle in a Haystack (Kamradt, 2023), a notable gap remains in the ability of LLMs to generate long aligned outputs effectively, as illustrated in Figure 1, and this issue cannot be easily addressed by manipulating decoding strategies. We analyze that this is due to a training gap: during the pre-training stage, despite having access to a vast array of long text sources, there is a lack of effective instructions to cultivate this capability. Conversely, in the post-training stage, the majority of human-conducted or AI-augmented query-response pairs are short, which leads to the trained LLMs facing challenges in generating long outputs.

To address this issue, current efforts employ two strategies: instruction backtranslation (Li et al., 2023) and behavior imitation (Xu et al., 2024), to construct post-training data with long responses. Instruction backtranslation leverages existing long-form, high-quality human-written texts, such as those found in magazines or books, as responses, and uses them to generate corresponding queries. However, obtaining high-quality data that spans various long-generation tasks and domains is challenging, and much of the available data presents risks of copyright infringement (Maini et al., 2024). On the other hand, behavior imitation aims to cost-effectively leverage proprietary models like GPT-4 to generate lengthy responses. This approach, which assumes the existence of a more proficient

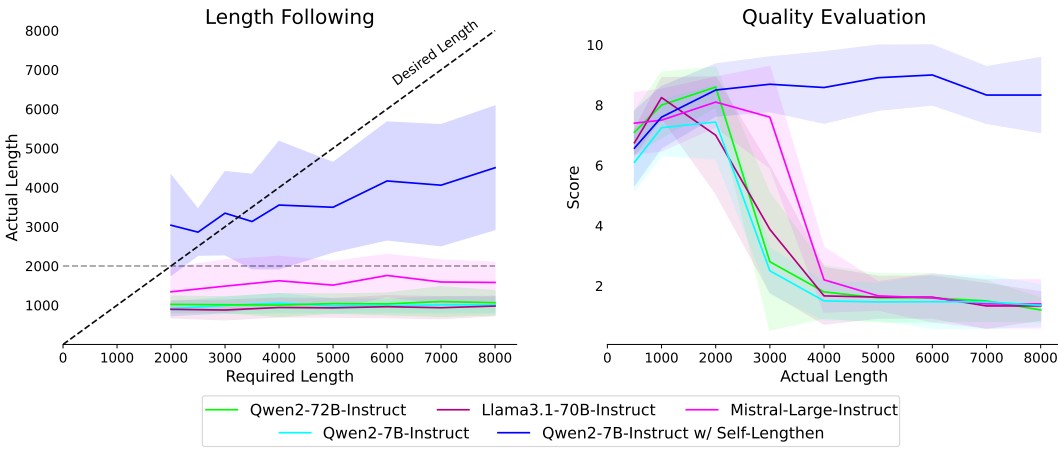

Figure 1: **Existing LLMs are struggling to generate long outputs.** (Left) The actual output lengths when prompting LLMs for generation tasks with specific length constraints under default parameters. All four tested LLMs struggle to exceed 2000 words. (Right) We attempted to get longer outputs by adjusting the decoding strategies (sampling and beam-search) and parameters (min tokens, length penalty, etc.). However, the quality evaluated by humans significantly deteriorated after reaching 2000 words. By employing our proposed Self-Lengthen method, we significantly enhanced the output length of the Qwen2-7B-Instruct backbone model while preserving the quality of the generated content.

model, is constrained by OpenAI's terms of use[1]. Additionally, we currently lack a clear understanding of how to build a model capable of generating long texts from scratch.

In this paper, for the first time, we demonstrate how we can stimulate the long-generation ability using only the LLM's intrinsic knowledge and skills, without the need for any auxiliary data or proprietary models. At the core of our approach is an iterative training framework called Self-Lengthen, which consists of two roles: the Generator and the Extender. The Generator is responsible for generating the initial response, while the Extender's task is to lengthen the response. Specifically, at the beginning, both the Generator and the Extender are initialized using an existing instruct model. Given a query, the Generator first outputs a response. Then, we split the response into two parts based on punctuation. The Extender is prompted to expand the first part, after which it combines the expanded version of the first part with the original response to revise and expand the second part. This procedure results in a new response that is approximately twice as long as the original. Next, the Generator is trained to directly produce this longer response based on the query, while the Extender is fine-tuned to extend the original response into this longer one. This results in updated versions of the Generator and Extender after each round of training. By iteratively applying this process, the Generator and Extender are progressively trained to handle increasingly longer responses. Once the Generator is able to produce outputs of the desired length, it can be used to generate long query-response pairs, effectively creating the post-training data needed for long-form tasks.

Experiments conducted on both benchmarks and human evaluations demonstrate that Self-Lengthen consistently achieves better long-text generation capabilities compared to instruction backtranslation and behavior imitation, even without the need for additional long-form text data or proprietary models. A more detailed analysis reveals that with each iteration of Self-Lengthen training, both the Generator and the Extender are able to produce content that is approximately twice as long as the output from the previous round, while maintaining the same level of quality.

## 2 RELATED WORK

**Long Text Generation** Most previous works explore long text generation in a hierarchical manner. Fan et al. (2018) initially propose to create a story by first generating a short summary and

---
[1]https://openai.com/policies/terms-of-use

then improve this method by introducing an intermediate step of producing the predicate-argument structure of the story as an outline (Fan et al., 2019). Tan et al. (2020), Sun et al. (2020) and Yao et al. (2019) further refine this hierarchical long text generation technique, followed by more advanced methods like Re[3] (Yang et al., 2022b) and its variant DOC (Yang et al., 2022a), which introduced a recursive prompting method for LLMs for long story generation in a plan-and-write fashion. Additionally, STORM (Shao et al., 2024) organizes long-form articles by first using retrieval and multi-perspective questions asking to develop an outline and then writing each section in parallel. Another series of studies employ human-in-the-loop approaches to interactively create long articles (Coenen et al., 2021; Chung et al., 2022; Goldfarb-Tarrant et al., 2019; Zhou et al., 2023).

**Long Output Alignment** While the above long-generation methods mainly target specific text types (*e.g.*, stories), recently there has been a growing interest in aligning LLMs to follow long output instructions. The latter is often more challenging due to the greater diversity in generation types. Weaver Wang et al. (2024) series is a set of close-source commercial models designed specifically for creative writing. Suri (Pham et al., 2024) gathers high-quality long-form human-written text to employ instruction backtranslation and demonstrates that fine-tuned models can effectively extend LLM output lengths. LongWriter broadens query types by adopting a plan-and-write pipeline and applying behavior imitation on GPT-4o to follow more diverse instructions. While existing methods primarily achieve long-form output based on existing texts or more powerful LLM, we explore a new paradigm that stimulates long-generation ability from scratch using only the LLM's intrinsic knowledge and skills by iteratively self-lengthen the output length and inductively self-align to generate increasingly longer outputs.

## 3 PREREQUISITION AND PRELIMINARIES

**Prerequisition** Compared with existing long output methods, our approach only requires a set of seed instructions on long output tasks and an open-source instruction model to automatically improve the model's ability to output long texts. A comparison of resource requirements is listed in Table 1. Our high-level idea is to use the existing models to longer its output in each round, and in turn fine-tune the existing models to have longer output capabilities. Then, we iterate this process to continuously construct longer data and more powerful long output models. To

| Method | Suri (Backtranslation) | LongWriter (Behavior Imitation) | Self-Lengthen (Self-Alignment) |
|---|---|---|---|
| Resource | High-quality Human-written text | Seed instructions | Seed instructions |
| LLM | Open-source instruct model | High-capability long-context LLM (*GPT-4 in their paper*) | Open-source instruct model |
| Output | Only cover a small range of tasks | Rigidly structured | All styles and types |

Table 1: Comparison of resources, LLM, and output for different long output data sourcing methods.

achieve this goal, two critical models: the Generator and the Extender, will be iteratively used and trained.

**Generator** Given an **instruction** $x$ guiding long output, our Generator can generate a **long output** $y$ within a certain length range. We will **iteratively train the Generator** $\text{Gen}_i$ in the $i$-th iteration, with each initialized from the previous iteration. In particular, at the beginning, $\text{Gen}_0$ is directly initialized from a well-aligned instruct model. During the iteration process, the output length of the Generator will **continuously increase**. Formally, we use $\text{Gen}_\theta$ to denote the Generator with parameters $\theta$, respectively. Then, the likelihood $\text{Gen}_\theta(y|x)$ for long output $y$ is obtained by cumulatively multiplying the conditional probability of each next token $p(y_t|x, y_{<t})$ as shown in Equation (1).

$$\text{Gen}_\theta(y|x) = \prod_{i=1}^{len(y)} p_\theta(y_i|x, y_{<i}) \tag{1}$$

**Extender** Our Extender can extend a long response $y$ to a **longer response** $y^+$ under its instruction $x$. Similar to the Generator series, we will train an Extender $\text{Ext}_i$ that can extend responses to a longer length based on the previous iteration, and the first round Extender $\text{Ext}_0$ is initialized from the seed instruction model with appropriate **extend prompt** $\text{prompt}_{EXT}$. The likelihood

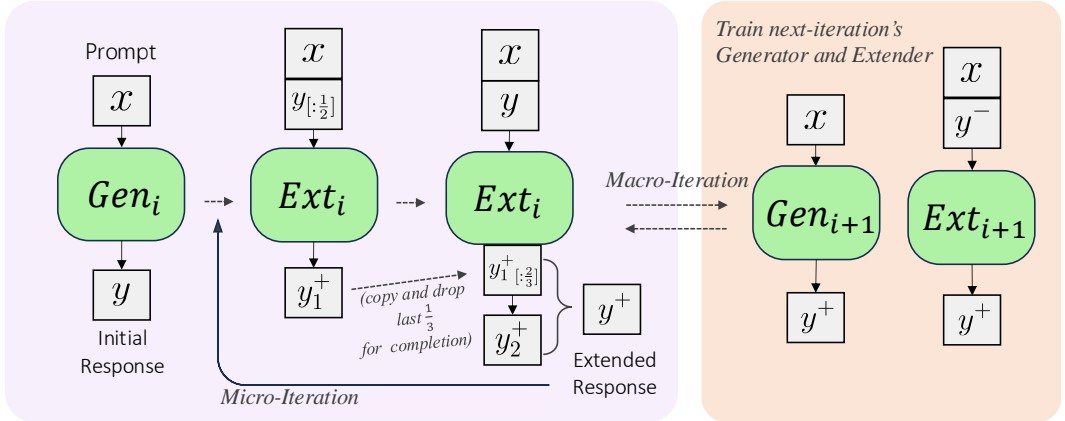

Figure 2: A high-level overview of the proposed Self-Lengthen. (Micro-Iteration) We begin by using the Generator to produce an initial response. Then, we employ the Extender to expand this response in a two-stage process, resulting in a much longer output. This extension process can be repeated iteratively to create increasingly lengthy responses. (Macro-Iteration) Once we have generated long responses, we use them to fine-tune both the Generator and Extender. These improved models are then utilized in subsequent iterations to generate and extend even longer responses.

$y^+ \sim \text{Ext}_\theta(y^+|\text{prompt}_{EXT}(x, y))$ for extended response $y^+$ can be represented as Equation (2).

$$\text{Ext}_\theta(y^+|x, y) = \prod_{i=1}^{len(y^+)} p_\theta^{EXT}(y^+{}_i|x, y, y^+{}_{<i})$$ (2)

Within each iteration, $len(y^+) > len(y)$ will always hold, and we will increase $len(y)$ and $len(y^+)$ during iterations.

# 4 METHODOLOGY: SELF-LENGTHEN

Based on user instructions, our algorithm empowers the seed instruct model to autonomously iterate to enhance the ability to produce longer outputs. This involves a process of gradually expanding the model's output while training it in reverse, effectively increasing the length and content of its generated output step by step. We show our high-level methodology in Figure 2.

## 4.1 INDUCTIVELY EXTENDING LONG OUTPUT ABILITY

The workflow of our Self-Lengthen can be split into four essential steps: 1) instruction augmentation, 2) initial response generation, 3) response extension, and 4) fine-tuning the next iteration's models. These four steps will be iteratively conducted and inductively extend the LLM's output length.

**Instruction Augmentation**   We employ self-instructing to bootstrap our instruction set. Specifically, we maintain an instruction pool and randomly select two instructions at a time as few-shot learning examples to generate a new instruction. This approach fosters deep exploration and a diverse array of instructions. Additionally, we employ the seed LLM to validate the generated instructions, filtering out any unsuitable instructions for producing long responses.

**Initial Response Generation**   In the $i$-th iteration, we employ the Generator $\text{Gen}_i$, which has been fine-tuned from the previous iteration, to produce initial responses $y$ based on the augmented instructions $x$. As the Generator's ability to generate long outputs improves with each iteration, the length of these initial responses will also grow progressively.

**Response Extension**   After generating the initial response $y$, the next step is to extend its length to create a longer response $y^+$. The vanilla extension method, namely directly using a prompt to

generate an extended response, is limited to the model's upper length constraint and cannot yield continuous increases during iterations. To overcome this, we propose a two-stage extension method that allows us to reach and exceed the model's length limits.

- In Stage 1, we use the first half of the initial response, denoted as $y_{[:\frac{1}{2}]}$, as input for the extension process, $\text{Ext}_i$. This yields an extended version $y_1^+$ for the first half of the content. Because the input is shorter, the model can produce a significantly longer output than the input—potentially expanding $y_{[:\frac{1}{2}]}$ to almost twice its original length, matching about $\text{len}(y)$.

- In Stage 2, we provide the entire initial response $y$ to $\text{Ext}_i$ for further expansion. Here, $y_1^+$ serves as the existing extended content for in-context learning (ICL), illustrating the structure and content of the extension. Since $y_1$ is derived from the first half of $y$, we anticipate that this approach will facilitate further expansion of the subsequent content in a similar manner.

A key technique used in this process is the removal of the last third of $y_1^+$, denoted as $y_{1\,[:\frac{2}{3}]}^+$. This ensures that the extension does not end properly, creating space for the model to seamlessly connect the preceding and following segments, thereby enhancing its ability to complete the expansion task. We don't need to identify which part of the original $y_1$ corresponds to the omitted third; we allow the model to handle this as part of its extension process. An illustration of this process is shown in Figure 3.

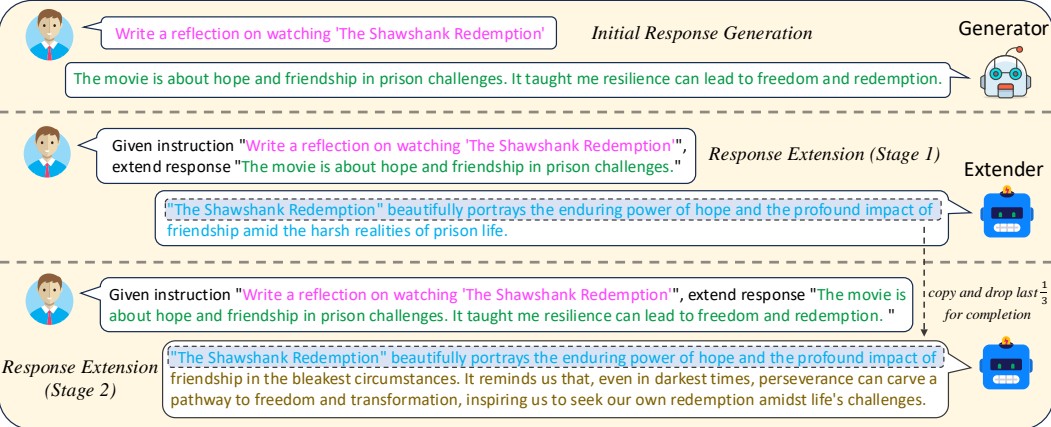

Figure 3: An illustration of the response extension process. Given an instruction, we first employ the Generator to produce an initial response. We then use the Extender to take the first half of this initial response to create the extension of the first part (*Stage 1*). Following this, we proceed to extend the entire response, using the previous two-thirds of the extended first half for in-context learning (ICL) to complete the extension of the remaining part (*Stage 2*). Ultimately, the model delivers a cohesive and consistent extended response. Note that this example is just for illustrative purposes since in real scenarios the responses would be much longer.

This extension step can be repeated multiple times within a single iteration, which we refer to as an inner-iteration. Specifically, after we apply a two-stage extension to transform the initial response $y$ into $y^+$, we can perform another two-stage extension by substituting the input $y$ with $y^+$ to obtain a further extended version $y^{++}$. We will apply this process three times within each iteration, retaining only those responses that are genuinely longer after the extensions.

**Fine-tuning the Next Iteration's Models**   Once we obtain the final extended responses $y^+$ (we use $y^+$ for simplicity, but it could also be $y^{++}$ or $y^{+++}$ depending on our inner iteration process), we will utilize these to fine-tune the Generators $\text{Gen}_{i+1}$ and Extenders $\text{Ext}_{i+1}$ for enhancing their capability to produce longer outputs. At this stage, we will also apply rule-based methods to filter out invalid responses to ensure their quality. Specifically, we will eliminate responses exhibiting any of the following issues:

1. **Inadequate length:** The length of the extended response does not exceed $120\%$ of the initial response length, hindering the depth of analysis.

2. **Frequent repetition:** The response excessively reiterates the same phrases or sentences.

3. **Endless:** The extended response does not end with normal punctuation.

4. **Code-switching:** The response incorporates unintended languages, *e.g.*, an English article containing inappropriate Chinese characters.

To accelerate the process of length increase during iterations, we implement a *length-bias sampling* technique. This involves randomly down-sampling responses that have shorter lengths. Specifically, we denote the set of extended responses as $S$ and define $r(len(y^+))$ as the length percentile of $y^+$ (the shortest response will be 0 and the longest will be 1). We then sample to create our length-biased set $S'$ as follows, where the shorter responses are assigned a much higher dropping rate:

$$S' = \{y^+ \in S : U(0,1) > 2 * (1 - r(len(y^+)))^3\} \tag{3}$$

After obtaining the pruned extended responses, we will perform supervised fine-tuning to get $\text{Gen}_{i+1}$ on $(x, y^+)$ pairs, with the parameters initializing from $\text{Gen}_i$ as a warm-up. For the Extender, we will first randomly remove 15% of the lines from the initial responses $y$ to create manipulated responses $y^-$. We will then use $(x, y^-, y^+)$ pairs, incorporated into $\text{prompt}_{EXT}$, to perform supervised fine-tuning on $\text{Ext}_i$ to obtain $\text{Ext}_{i+1}$. The random dropping of lines encourages the Extender to complete critical missing information within paragraphs, thereby enhancing its extending capabilities.

## 4.2 Final Alignment

Our goal isn't just to acquire a specific Generator or Extender; we aim to enhance the long text generation capabilities of any seed model while maintaining its overall performance. To accomplish this, after multiple iterations of our method, we will gather query-response pairs and implement length controls in the queries to create SFT data.

**Data Collecting** We will gather data from the initial responses $y$ and extended responses $y^+$, along with their corresponding prompts $x$, in each iteration to create a more extensive dataset. This dataset will encompass a variety of prompt lengths and types. Additionally, we will use the Generators in each iteration to generate more data, further enriching our dataset.

**Query Rephrasing for Length Control** In our earlier steps, we did not sufficiently highlight the importance of length control in the input instructions. As a result, the Generators produce responses of varying lengths for the same instruction across different iterations. To develop a general model, we will now incorporate length constraints in the input instructions. Specifically, we will first count the length of each response, denoted as $len(y)$, and then utilize our seed LLM to integrate this constraint into the input instructions. We will strategically design prompts to create a diverse range of output instructions that encompass various types of length restrictions.

## 5 Experiment and Evaluation

To demonstrate the effectiveness of our method, we chose two other mainstream methods as baselines for comparison:

- **Backtranslation:** We compare our method with instruction backtranslation from high-quality human-written text, which has been demonstrated to be effective in extending models' output length, according to Suri (Pham et al., 2024). We randomly selected 3,000 samples from the Suri SFT dataset as our English text source and collected 3,000 high-quality Chinese SFT data ranging from 2,000 to 8,000 in length from the web to create our Chinese dataset.

- **Plan-and-Write:** We compare our method with the hierarchical plan-and-write methods, which have been widely studied in previous research. Among them, we select LongWriter (Bai et al., 2024) as our basic implementation since it can handle a broader range of query types and is more suitable for our experiment settings. We source the data through 1) behavior imitation using GPT-4o, and 2) self-alignment using Qwen2-7B-Instruct.

We will evaluate our method against these baselines in two key aspects: 1) generated data and 2) fine-tuned models.

## 5.1 DATA EVALUATION

To conduct a comprehensive assessment, we carried out a human evaluation of the generated data. In this evaluation, we focused mainly on comparing our method with backtranslation and hierarchical plan-and-write approaches. To control for variables, we centered our analysis on the generated data rather than the finely-tuned model. We randomly selected 15 query-response pairs from each method, ensuring that their lengths were approximately uniformly distributed between 2k and 8k words.

For the queries, evaluators scored them based on how well they addressed user needs, their diversity, and their naturalness. When assessing responses, evaluators considered several factors to determine an overall satisfaction score, including relevance, coherence, accuracy, consistency, clarity, creativity, and engagement. Due to the extraordinarily long length of the responses, we utilized GPT-4o to summarize and analyze them, aiding evaluators in their assessments for improved readability and increased annotation efficiency. Both queries and responses were rated on a scale from 1 to 10. We recruited multiple individuals to annotate the data and record the average scores. Detailed evaluation guidelines can be found in the Appendix F.1.

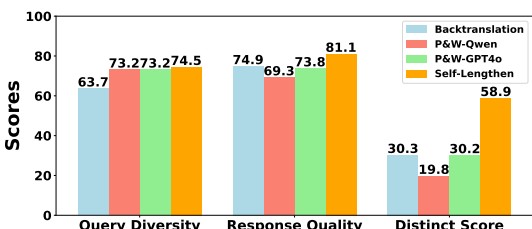

Figure 4: The results of data evaluation.

Additionally, we calculate the distinct scores of the generated responses and record the average, which measures response diversity. This metric is crucial for our long output generation tasks, especially those requiring creative writing. We have summarized all results in Figure 4 and identified two noteworthy findings.

Firstly, unlike Suri, which relies on instruction-backtranslations from various human-written texts, we utilize user logs, resulting in significantly better query diversity compared to backtranslation methods. We also found that the quality of our responses surpasses that of the backtranslation method created by humans and the behavior imitation method using GPT-4o. Both using Qwen2-7B-Instruct for data generation, our approach yields substantially higher response scores than Plan-and-Write-Qwen, highlighting the advantages of our method. Secondly, we observed a notable improvement over other methods in terms of distinct scores, indicating a wider variety of expressions and content. This finding provides an intuitive signal of our superior performance.

## 5.2 FINE-TUNED MODEL EVALUATION

**LonGen Benchmark and Metrics** In our evaluation, we collect and assemble a set of test prompts from our online logs. These prompts are meticulously tested to ensure they do not appear in our training set and do not contain any personally identifiable information (PII). The queries are very diverse and cover a wide range of real user needs across different long-form generation tasks. To protect user privacy when open-sourcing these prompts, we also utilize GPT-4o to rewrite them. The rewritten prompts adhere to strict length constraints, sourcing the LonGen benchmark, with detailed statics shown in Table 3. We then assess the responses based on two criteria:

1. **Length Following Score:** We categorize the length constraints into four groups: 1) the length is **about** a specific length, 2) the length falls within a specific **range**, 3) the length is **above** a specific length, and 4) the length is **below** a specific length. Based on the ground truth length defined in the queries and the model's actual output length, we calculate a scalar length-following score ranging from 0 to 1, where a higher score indicates greater adherence to the desired length. The detailed calculation method is provided in the appendix.

2. **Output Quality Score:** We employ LLM-as-a-judge to evaluate the quality of the generated responses. Specifically, we use GPT-4o to assess the responses on seven key aspects pertinent to long-form generation tasks: relevance, coherence, accuracy, consistency, clarity, creativity, and

Table 2: The main results on the LonGen. $S_L$ and $S_Q$ stand for the length-following score and response quality score, respectively. The highest score among different fine-tuning methods on the backbone model is highlighted in green.

| Model | Overall | | [2k, 4k) | | [4k, 6k) | | [6k, 8k) | |
|---|---|---|---|---|---|---|---|---|
| | $S_L$ | $S_Q$ | $S_L$ | $S_Q$ | $S_L$ | $S_Q$ | $S_L$ | $S_Q$ |
| *Open-source LLM* | | | | | | | | |
| Qwen2-72B-Instruct | 6.08 | 82.26 | 15.12 | 83.93 | 0.25 | 82.45 | 2.86 | 80.41 |
| Llama-3.1-70B-Instruct | 2.08 | 84.82 | 6.24 | 84.84 | 0.00 | 85.05 | 0.00 | 84.55 |
| Mistral-Large-Instruct | 15.30 | 86.05 | 39.07 | 87.02 | 6.83 | 85.70 | 0.00 | 85.43 |
| *Proprietary LLM* | | | | | | | | |
| GPT-4o | 14.98 | 85.75 | 41.00 | 85.41 | 3.62 | 85.59 | 0.31 | 86.25 |
| Claude-3.5-Sonnet | 46.01 | 85.34 | 71.14 | 85.68 | 43.81 | 85.30 | 23.08 | 85.04 |
| *Qwen Backbone* | | | | | | | | |
| Qwen2-7B-Instruct | 3.38 | 81.74 | 10.15 | 83.21 | 0.00 | 81.98 | 0.00 | 80.02 |
| w/ Backtranslation | 58.29 | 85.72 | 68.26 | 85.93 | 62.90 | 85.91 | 43.70 | 85.32 |
| w/ Plan-and-Write | 58.51 | 85.52 | 68.40 | 85.14 | 60.42 | 85.30 | 46.72 | 86.12 |
| w/ Self-Lengthen | 60.48 | 85.82 | 71.65 | 86.55 | 62.71 | 85.27 | 47.07 | 85.63 |
| *LLaMA Backbone* | | | | | | | | |
| Llama3.1-8B-Instruct | 2.36 | 79.40 | 7.08 | 81.70 | 0.00 | 78.68 | 0.00 | 77.85 |
| w/ Backtranslation | 18.50 | 43.03 | 9.56 | 39.29 | 10.32 | 44.76 | 35.63 | 45.43 |
| w/ Plan-and-Write | 49.36 | 82.86 | 60.70 | 81.43 | 59.33 | 87.14 | 28.05 | 79.76 |
| w/ Self-Lengthen | 51.61 | 83.43 | 62.07 | 80.00 | 60.61 | 84.29 | 32.15 | 84.90 |

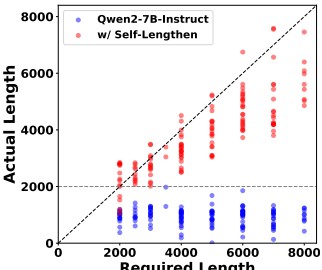

**# Data in each subset**

| Language | | | |
|---|---|---|---|
| Chinese | | | 120 |
| English | | | 120 |
| **Type** | [2k, 4k) | [4k, 6k) | [6k, 8k) |
| about | 10 | 10 | 10 |
| range | 10 | 10 | 10 |
| above | 10 | 10 | 10 |
| below | 10 | 10 | 10 |

| Output Category | |
|---|---|
| Literature and Creative Writing | 66 |
| Academic Research | 64 |
| Business Communication | 37 |
| Journalism and Media | 26 |
| Miscellaneous Professional Writing | 18 |
| Personal Expression | 15 |
| Technical Documentation | 14 |

Table 3: Key statistics of LonGen, which contains 2 languages * 3 length ranges * 4 constraint types * 10 = 240 pieces of data in total.

Figure 5: Required-actual length scatters on LonGen.

engagement. Each aspect will be assigned a scalar score ranging from 1 to 10. We then calculate the average score across all aspects to arrive at the final quality score. The judging template is included in the appendix.

In our evaluation, we begin by testing the performance of several widely-used open-source and proprietary LLMs. We then use Qwen2-7B-Instruct as our backbone model and evaluate the performance of fine-tuned models using baseline methods and ours. Notably, the original versions of Suri and LongWriter datasets either lack appropriate length requirements in their queries or have unsuitable length constraints, resulting in results that fall significantly short of expectations. To ensure a fair evaluation, we introduce length constraints by applying the same workflow and prompts to their original queries. Finally, we compare the improvements achieved against the backbone LLM.

The results are presented in Table 2. We observe that most existing open-source and proprietary LLMs struggle to generate outputs longer than 4,000 words, consistent with our preliminary study. However, after fine-tuning on our sourced dataset, we achieved a significant increase in output length compared to general models. This improvement is also advantageous when compared to backtranslation and the plan-and-write approaches. Furthermore, we find that the quality of the outputs outperforms the other two methods overall and surpasses that of the backbone model by a considerable margin. The only aspect that is reduced is clarity, which is understandable, as longer outputs tend to increase the read complexity, especially when the content is much richer.

These results demonstrate our effectiveness of employing a relatively small LLM that self-aligns for competitive long-output capabilities, even exceeding those of much larger LLMs and other long-output methods that require human-written text or high-capability long-context LLMs.

**Self-Lengthen Exhibits Strong Performance in GPT-4o Pair Comparisons** We leveraged GPT-4o to evaluate pairs of responses generated by various fine-tuned models in response to the same queries from our evaluation set. The win rate heat map is displayed in Figure 6. These results revealed that all specialized models demonstrated significant improvements over the backbone model in long-generation tasks. Furthermore, our methods surpassed both the instruction backtranslation and plan-and-write approaches.

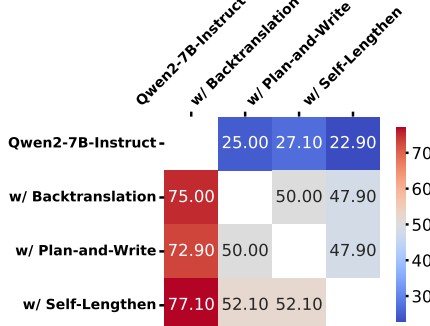

Figure 6: The win-rate heat map.

**Self-Lengthen Enables a Steady Increase in Output Length across Iterations** At each iteration, we assess the generated output length and the associated scores as evaluated by GPT-4o, and display their distribution in Figure 7. We document both the initial responses and the final extended responses, designating these as results for the generator and extender, respectively. Additionally, we track the performance of the seed model as iteration 0. The figure demonstrates that both the generator and extender models experience a consistent increase in average output length with each iteration, and we will extend the length to approximately twice its original size in each iteration. We also observe that while longer outputs may be accompanied by a slight decrease in scores, this drop is negligible and can be disregarded throughout the iterations.

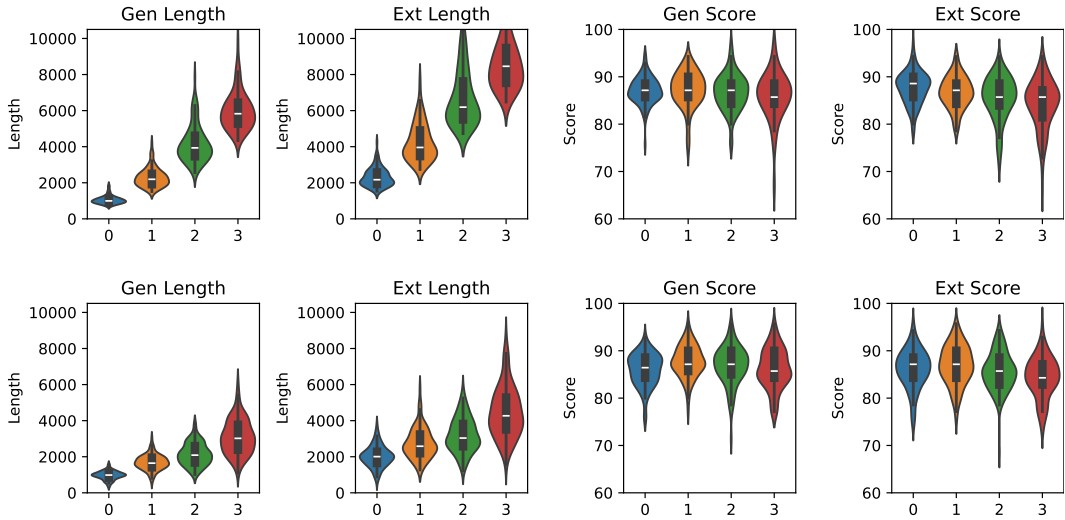

Figure 7: The length and score distribution in each iteration. Compared with not using length bias sampling (Lower Row), we find that using length bias sampling (Upper Row) will significantly speed up length extending process, and do not have signs to aggravate the score dropping.

**The Output Styles can be Simply Adjusted via Changing the Extend Prompt** We have discovered that modifying the extended prompt allows us to alter the final output style, and the results are sensitive to these adjustments. This finding highlights the flexibility and significant potential of our method. For instance, if we wish to make the generated content more narrative rather than technical, we can simply emphasize that point in the prompt. This approach enables us to customize the output according to our preferences or to achieve various text styles by tweaking the prompt.

To illustrate this finding, we enhanced the prompts by attaching: 1) "You should elaborate on the details to make each paragraph richer" and 2) "You should add more examples and subparagraphs" while keeping all other components unchanged. We then conducted three macro-iterations for both extend prompts and selected 1,000 samples of length between 4,000 and 6,000 words to analyze the number of paragraphs produced. We present the results in Figure 8. We found that extend prompt 2 exhibits a significant rightward shift in paragraph count distribution compared to extend prompt 1, with an average increase of 10% in the number of paragraphs. This suggests a change in its output style.

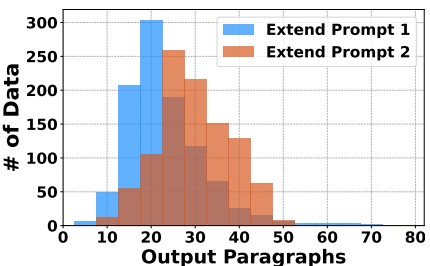

Figure 8: The distributions of paragraph counts using two different extending prompts.

## 6 CONCLUSION

In this work, we introduced Self-Lengthen, which demonstrates for the first time how to elicit high-quality, length-following responses from existing LLMs without relying on additional data or proprietary models. Self-Lengthen alternately trains a Generator and an Extender, where the Extender incrementally expands the Generator's responses in segments, serving as a training objective to enhance both the length of the Generator's subsequent response and the Extender's subsequent extension. Through three macro iterations, we can produce responses that are about eight times longer, thereby constructing high-quality (query, long response) training data. Our method has been rigorously evaluated using both benchmark metrics and human assessments, which have consistently confirmed the quality of the training data produced by Self-Lengthenand the effectiveness of the trained models.

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

## A  BOARDER IMPACT & LIMITATION

**Boarder Impart**  Given the flexibility of our extend template, we can apply this workflow to various tasks beyond its initial scope. For example, we can use it to enhance reasoning paths or Chain of Thought (CoT) processes, potentially leading to more robust inference time scaling. Specifically, this workflow can be employed to enrich existing problem solutions, allowing for more comprehensive and nuanced approaches.

**Limitation**  Due to resource constraints, our experiments were limited to language models with less than 10B parameters. We anticipate that our pipeline will be even more effective with larger LLMs, and we plan to validate this property in the future.

## B  MODEL CARDS & GENERAL BENCHMARKS

We list the details of our evaluated models in Table 4.

| Model name | Model version | Context window | Max output tokens |
|---|---|---|---|
| Claude 3.5 Sonnet (Anthropic, 2024) | claude-3-5-sonnet-20240620 | 200,000 tokens | 4,096 tokens |
| GPT-4o (Achiam et al., 2023) | gpt-4o-2024-08-06 | 128,000 tokens | 4,096 tokens |
| Qwen2-7B-Instruct (Yang et al., 2024) | - | 128,000 tokens | - |
| Qwen2-72B-Instruct (Yang et al., 2024) | - | 128,000 tokens | - |
| Llama-3.1-8B-Instruct (Dubey et al., 2024) | - | 128,000 tokens | - |
| Llama-3.1-70B-Instruct (Dubey et al., 2024) | - | 128,000 tokens | - |
| Mistral-Large-Instruct (Jiang et al., 2023) | Mistral-Large-Instruct-2407 | 128,000 tokens | - |

Table 4: Model cards.

## C  MORE RESULTS

We show the detailed length following evaluation results in Table 5 and the detailed response quality evaluation results in Table 6.

Table 5: The detailed results of the length following score on the LonGen. The four different length constraints about, range, above, and below are represented by $\approx$, $[]$, $>$, and $<$, respectively. The highest score among different fine-tuning methods on the backbone model is highlighted in green.

| Model | Overall | | | | [2k, 4k) | | | | [4k, 6k) | | | | [6k, 8k] | | | |
|---|---|---|---|---|---|---|---|---|---|---|---|---|---|---|---|---|
| | $\approx$ | $[]$ | $>$ | $<$ | $\approx$ | $[]$ | $>$ | $<$ | $\approx$ | $[]$ | $>$ | $<$ | $\approx$ | $[]$ | $>$ | $<$ |
| *Open-source LLM* | | | | | | | | | | | | | | | | |
| Qwen2-72B-Instruct | 3.75 | 4.78 | 4.02 | 11.76 | 11.24 | 9.73 | 10.04 | 29.45 | 0.00 | 0.00 | 0.00 | 1.00 | 0.00 | 4.60 | 2.01 | 4.82 |
| Llama-3.1-70B-Instruct | 1.14 | 0.65 | 1.86 | 4.67 | 3.42 | 1.95 | 5.59 | 14.01 | 0.00 | 0.00 | 0.00 | 0.00 | 0.00 | 0.00 | 0.00 | 0.00 |
| Mistral-Large-Instruct | 14.44 | 12.50 | 14.64 | 19.63 | 35.15 | 33.63 | 34.19 | 53.33 | 8.16 | 3.88 | 9.74 | 5.56 | 0.00 | 0.00 | 0.00 | 0.00 |
| *Proprietary LLM* | | | | | | | | | | | | | | | | |
| GPT-4o | 13.19 | 12.78 | 16.23 | 17.71 | 36.79 | 36.44 | 42.10 | 48.67 | 2.77 | 1.84 | 6.15 | 3.70 | 0.00 | 0.05 | 0.43 | 0.75 |
| Claude-3.5-Sonnet | 47.02 | 42.09 | 54.43 | 40.50 | 69.80 | 69.22 | 76.02 | 69.52 | 46.77 | 31.88 | 61.42 | 35.16 | 24.49 | 25.17 | 25.84 | 16.84 |
| *Qwen Backbone* | | | | | | | | | | | | | | | | |
| Qwen2-7B-Instruct | 1.54 | 0.95 | 0.46 | 10.59 | 4.63 | 2.84 | 1.37 | 31.77 | 0.00 | 0.00 | 0.00 | 0.00 | 0.00 | 0.00 | 0.00 | 0.00 |
| w/ Backtranslation | 62.08 | 45.48 | 52.71 | 72.89 | 68.24 | 66.09 | 73.96 | 64.76 | 74.16 | 38.38 | 57.74 | 81.31 | 43.85 | 31.96 | 26.41 | 72.59 |
| w/ Plan-and-Write | 60.94 | 44.32 | 54.93 | 73.87 | 78.81 | 64.65 | 69.37 | 60.77 | 67.37 | 37.26 | 53.33 | 83.71 | 36.63 | 31.04 | 42.10 | 77.12 |
| w/ Self-Lengthen | 61.63 | 51.90 | 52.07 | 76.32 | 73.45 | 86.32 | 69.13 | 57.72 | 67.42 | 33.36 | 56.30 | 93.75 | 44.01 | 36.04 | 30.77 | 77.48 |
| *LLaMA Backbone* | | | | | | | | | | | | | | | | |
| Llama3.1-8B-Instruct | 1.68 | 0.00 | 0.82 | 6.94 | 5.03 | 0.00 | 2.45 | 20.83 | 0.00 | 0.00 | 0.00 | 0.00 | 0.00 | 0.00 | 0.00 | 0.00 |
| w/ Backtranslation | 13.23 | 12.91 | 26.88 | 20.98 | 0.57 | 6.53 | 4.20 | 26.93 | 5.37 | 4.28 | 20.73 | 10.88 | 33.75 | 27.92 | 55.71 | 25.13 |
| w/ Plan-and-Write | 55.82 | 34.60 | 44.48 | 62.55 | 70.80 | 61.61 | 63.15 | 47.26 | 72.00 | 27.65 | 54.26 | 83.38 | 24.65 | 14.53 | 16.02 | 57.01 |
| w/ Self-Lengthen | 44.48 | 44.28 | 44.59 | 73.09 | 51.22 | 71.24 | 56.30 | 69.52 | 59.11 | 46.85 | 57.30 | 79.19 | 23.10 | 14.74 | 20.17 | 70.58 |

Table 6: The detailed results of the response quality score on the LonGen. The highest score among different fine-tuning methods on the backbone model is highlighted in green.

| Model | Overall | Relevance | Coherence | Accuracy | Consistency | Clarity | Creativity | Engagement |
|---|---|---|---|---|---|---|---|---|
| *Open-source LLM* | | | | | | | | |
| Qwen2-72B-Instruct | 82.26 | 90.67 | 86.12 | 87.00 | 89.92 | 87.33 | 65.29 | 69.50 |
| Llama-3.1-70B-Instruct | 84.82 | 95.38 | 88.62 | 88.92 | 92.54 | 88.21 | 67.50 | 72.54 |
| Mistral-Large-Instruct | 86.05 | 96.54 | 89.38 | 91.29 | 93.42 | 88.25 | 69.42 | 74.04 |
| *Proprietary LLM* | | | | | | | | |
| GPT-4o | 85.75 | 97.21 | 89.00 | 90.71 | 93.29 | 88.79 | 68.17 | 73.08 |
| Claude-3.5-Sonnet | 85.34 | 95.79 | 89.00 | 90.25 | 93.08 | 86.04 | 70.46 | 72.75 |
| *Qwen Backbone* | | | | | | | | |
| Qwen2-7B-Instruct | 81.74 | 90.88 | 85.42 | 86.25 | 89.75 | 87.00 | 64.25 | 68.62 |
| w/ Backtranslation | $85.72_{+3.98}$ | $96.67_{+5.79}$ | $88.67_{+3.25}$ | $90.67_{+4.42}$ | $92.50_{+2.75}$ | $85.08_{-1.92}$ | $72.46_{+8.21}$ | $74.00_{+5.38}$ |
| w/ Plan-and-Write | $85.52_{+3.78}$ | $96.25_{+5.37}$ | $88.04_{+2.62}$ | $89.62_{+3.37}$ | $92.29_{+2.54}$ | $84.96_{-2.04}$ | $73.08_{+8.83}$ | $74.42_{+5.80}$ |
| w/ Self-Lengthen | $85.82_{+4.08}$ | $96.21_{+5.33}$ | $88.54_{+3.12}$ | $89.88_{+3.63}$ | $92.71_{+2.96}$ | $84.71_{-2.29}$ | $73.21_{+8.96}$ | $75.46_{+6.84}$ |
| *LLaMA Backbone* | | | | | | | | |
| Llama3.1-8B-Instruct | 79.40 | 88.02 | 83.63 | 84.51 | 87.17 | 84.64 | 61.69 | 66.12 |
| w/ Backtranslation | $43.03_{-36.37}$ | $54.12_{-33.90}$ | $38.24_{-45.39}$ | $68.24_{-16.27}$ | $50.00_{-37.17}$ | $43.53_{-41.11}$ | $23.53_{-38.16}$ | $23.53_{-42.59}$ |
| w/ Plan-and-Write | $82.86_{+3.46}$ | $93.53_{+5.51}$ | $87.06_{+3.43}$ | $86.47_{+1.96}$ | $90.59_{+2.54}$ | $82.94_{-1.70}$ | $67.65_{+5.96}$ | $71.76_{+5.64}$ |
| w/ Self-Lengthen | $83.43_{+4.03}$ | $95.33_{+7.31}$ | $87.33_{+3.70}$ | $90.67_{+6.16}$ | $94.00_{+8.63}$ | $79.33_{-5.31}$ | $68.00_{+6.31}$ | $69.33_{+3.21}$ |

# D  IMPLEMENTATION DETAILS

We conduct experiments using Qwen2-7B-Instruct and Llama3.1-8B-Instruct as the backbone models across multiple nodes equipped with Nvidia H100 GPUs and Intel® Xeon® Processors. For inference, we utilize FastChat (Zheng et al., 2023) to control vLLM (Kwon et al., 2023) workers for high throughput. During training, we use Llama-Factory (Zheng et al., 2024) as the high-level API and implement DeepSpeed ZeRO-3 (Rajbhandari et al., 2020) and CPU offloading to accelerate the training process while minimizing GPU memory consumption.

# E  PILOT STUDY

In our pilot study, we aim to investigate the long-output capabilities of existing models. We select ten queries suitable for eliciting long responses and adjusted the length requirements within these queries to generate outputs of varying lengths, all while using default decoding parameters. We then document both the actual output lengths and the specified length demands.

To further ascertain whether adjusting decoding strategies can produce valid long outputs, we randomly sample various key decoding parameters within their permissible ranges and attempt to generate longer responses. Specifically, we test both sampling and beam-search decoding strategies, using randomly selected parameters outlined in Table 7. We also set the *min_tokens* to a sufficiently large number to force the models to generate adequately lengthy outputs.

|  | **Sampling** | **Beam-Search** |
|---|:---:|:---:|
| **length_penalty** | 1 | $[0, 2]$ |
| **repetition_penalty** | $[0, 2]$ | $[0, 2]$ |
| **temperature** | $[0, 2]$ | 0 |
| **top_p** | $[0, 1]$ | 1 |
| **best_of** | 1 | $[2, 10]$ |

Table 7: Sampling range for decoding parameters across two strategies. We will randomly choose decoding strategies and sample the appropriate parameters to generate extended outputs. Subsequently, we will evaluate the quality of responses of suitable lengths using GPT-4o, selecting only the top 1% for human evaluation.

# F  EVALUATION DETAILS

## F.1  HUMAN EVALUATION GUIDANCE

For each method, we provide all queries from the randomly selected data at once for the annotator to assess the query scores. The evaluation instructions are displayed in the text box below.

> **Instruction of evaluating queries for human**
>
> Please assess the following user queries for generating long responses. Evaluate how well these queries meet user needs, their diversity, and overall naturalness. Then, assign an overall score from 1 to 10, with 1 being the lowest and 10 the highest.

To enhance readability and improve annotation efficiency given the lengthy nature of the responses, we first utilize GPT-4o to summarize the responses and provide a brief analysis to aid evaluators in their assessments. We present the prompt used for this process below.

> **Prompt to summarize and analyze the long responses for GPT-4o**
>
> You are a meticulous and impartial analysis assistant. Given a user prompt and the corresponding lengthy response, please first summarize the response in approximately 400 words. Then, provide a straightforward analysis of the quality of the response, including both strengths and weaknesses.
> User prompt: query
> Long response: response

After that, we direct the evaluators to access each response using the following instructions.

> **Instruction of evaluating responses for human**
>
> Please assess your satisfaction with the model's response under the user's query. Consider factors such as relevance, logic, accuracy, consistency, clarity, originality, and engagement, and give a score ranging from low to high [1, 10]. In your evaluation, you may refer to summaries and brief analyses provided by third-party models, but please note that this information is for reference only and should not influence your independent judgment.

For the evaluation of queries and responses, we also provide more detailed instructions for each score and several cases with analysis to guide the evaluators. However, we do not plan to display these instructions on the paper to conserve space.

## F.2 AUTOMATIC EVALUATION

### F.2.1 LENGTH FOLLOWING SCORE

For length constraints, we identify four types: 1) about, 2) range, 3) above, and 4) below. Based on these different types, we will calculate the corresponding target lower bound length $target_{min}$ and the target upper bound length $target_{max}$ as Table 8:

| type | constraint | target_min | target_max |
|------|------------|------------|------------|
| about | $\approx x$ | $0.8x$ | $1.2x$ |
| range | $[x_1, x_2]$ | $x_1$ | $x_2$ |
| above | $> x$ | $x$ | $1.5x$ |
| below | $< x$ | $0.5x$ | $x$ |

Table 8: The target minimum and maximum lengths for the four length constraint types.

Next, let $y$ represent the actual output length, and the length following score $S_L$ will be:

$$S_L = \begin{cases} 1 & if \, target_{min} \leq y \leq target_{max} \\ \max\left(0, \frac{y}{target_{min}} \cdot 2 - 1\right) & if \, y < target_{min} \\ \max\left(0, 3 - \frac{y}{target_{max}} \cdot 2\right) & if \, y > target_{max} \end{cases}$$

## G PROMPT TEMPLATE

We display the prompt templates used in our experiments. For self-instructing, we use the following template.

> **Prompt template for self-instructing**
>
> Your task is to construct user instructions that can guide the generation of long text.
>
> Instruction 1: {prompt1}
> Instruction 2: {prompt2}
> Instruction 3:

The prompt1 and prompt2 will be replaced with two randomly selected examples.

For response extending, we use the following template.

> **Prompt template for response extending**
>
> You are an expert in writing polishing. Based on the provided instruction and original outline, enhance the content by carefully expanding the original outline to increase its informational content,

incorporate specific details, and improve its descriptive quality. Remember that your task is not to continue writing content, but to enrich the current content. You can achieve this by enriching existing paragraphs with more detailed descriptions or creating more subheadings and sections. You should try to be creative and informative as much as possible and do NOT include any repetitive content or lengthy narratives in your response. You need to generate text that is as long, rich, and diverse as possible.

Instruction: {prompt}

Original outline:

{initial_response}

## H    CASE DEMONSTRATION

We present a case study to clarify our workflow. To start, we employ augmented instructions and use the generator to create an initial response.

### Instruction

Write a story about a young boy who discovers a magical book in his attic and learns how to harness the power of magic within himself.

### Initial response (553 words)

**The Boy Who Unlocked His Inner Magic**

Twelve-year-old Max had always been fascinated by the dusty, cobweb-covered trunks and boxes that filled his family's attic. He loved exploring the hidden treasures he found there, from old toys and clothes to mysterious artifacts and forgotten heirlooms. But nothing could have prepared him for what he was about to discover.

One sunny Saturday morning, while rummaging through an old trunk, Max stumbled upon a peculiar-looking book bound in worn leather. The cover was adorned with intricate symbols and patterns that seemed to shimmer in the dim light of the attic. As he opened the book, a faint hum filled the air, and the pages crackled with an otherworldly energy.

Intrigued, Max carefully flipped through the yellowed pages, revealing illustrations of fantastical creatures, magical spells, and ancient incantations. A note on the first page caught his eye:

> "For those who seek the spark within, To unlock the secrets of the heart, Read these words, and listen well,
> For the magic lies within you, not in this spell."

As Max read the words, he felt an unusual tingling sensation in his fingers, as if the book was awakening something deep within him. Suddenly, visions began to flood his mind – vivid images of enchanted forests, shimmering stars, and majestic dragons soaring through the skies.

Over the next few days, Max returned to the attic again and again, devouring the knowledge contained within the book. He practiced the simple spells and incantations, feeling the power coursing through his veins like electricity. At first, it was just a gentle buzzing, but soon the sensations grew stronger, and Max started to notice changes within himself.

He discovered he could make objects move with a mere thought, levitate small items, and even conjure tiny sparks of light from thin air. His friends at school noticed the change in him too; they'd see him effortlessly solve complex math problems or create beautiful, intricate drawings without any apparent effort.

But most remarkable of all, Max realized that the true source of his newfound abilities lay not in the book, but within himself. The book had simply unlocked a door to his own inner potential, allowing him to tap into a reservoir of untapped magic that had been slumbering inside him all along.

As Max continued to hone his skills, he learned to control his powers more effectively, mastering techniques to heal injuries, communicate with animals, and even bend time and space to his will. His confidence soared, and he knew that he was no longer just a ordinary boy; he was a wizard, capable of achieving incredible feats.

With great power comes great responsibility, however. Max understood that he must use his gifts wisely, helping others and protecting the world from those who would misuse such powerful forces. And so, with the magical book safely tucked away, Max embarked on a journey to explore the limits of his abilities, knowing that the true magic lay not in the pages of a book, but in the depths of his own heart and soul.

From that day forward, Max lived a life of wonder, using his powers to inspire and protect those around him, becoming a beacon of hope and a shining example of the magic that lay within every individual, waiting to be unleashed.

Next, we expand on the first half of the initial response and obtain the following content.

---

**Extended content on the first half response**

**The Boy Who Unlocked His Inner Magic**

**A World of Discovery**

Twelve-year-old Max had always been captivated by the dusty, cobweb-covered trunks and boxes that filled his family's attic. For years, he had spent countless hours exploring the hidden treasures he found there, from vintage toys and antique clothes to mysterious artifacts and forgotten heirlooms. Each new discovery sparked a sense of wonder and curiosity within him, fueling his imagination and inspiring his love for adventure. His parents often joked that he had a knack for uncovering the most unlikely treasures, and they would smile knowingly whenever he emerged from the attic with some newfound gem.

But nothing could have prepared Max for what he was about to discover on that fateful Saturday morning. As he climbed the creaky stairs leading up to the attic, the warm sunlight streaming through the windows illuminated the space, casting a golden glow over the familiar landscape of trunks, boxes, and forgotten relics. With a spring in his step, Max set out to explore the attic once again, determined to uncover a treasure that would surpass all others.

**The Mysterious Book**

While rummaging through an old trunk, Max's fingers brushed against a peculiar-looking book bound in worn leather. The cover was adorned with intricate symbols and patterns that seemed to shimmer in the dim light of the attic, like a constellation of tiny stars. As he opened the book, a faint hum filled the air, and the pages crackled with an otherworldly energy. The musty scent of aged paper wafted up, carrying with it whispers of secrets and mysteries long forgotten. Intrigued, Max carefully turned the pages, revealing illustrations of fantastical creatures, magical spells, and ancient incantations that seemed to leap off the page and dance before his eyes.

A note on the first page caught his eye, written in elegant script that seemed to shimmer like moonlight:

> "For those who seek the spark within, To unlock the secrets of the heart, Read these words, and listen well,
> For the magic lies within you, not in this spell."

As Max read the words, he felt an unusual tingling sensation in his fingers, as if the book was awakening something deep within him. It was as if the very essence of the book was resonating with his own heartbeat, calling forth a part of him that he never knew existed. Suddenly, visions began to flood his mind – vivid images of enchanted forests, shimmering stars, and majestic dragons soaring through the skies. The world around him melted away, leaving only the thrumming energy of the book and the promise of untold wonders waiting to be discovered.

**Unlocking the Secrets Within**

As Max delved deeper into the book, he began to realize that the words were not just mere ink on paper, but a key to unlocking the secrets of his own heart. The illustrations seemed to come alive, swirling with colors and textures that danced across the page like a kaleidoscope of dreams. The magical spells and incantations whispered secrets in his ear, guiding him toward a path of self-discovery and empowerment. With each passing moment, Max felt his connection to the book growing stronger, as if the very fabric of reality was weaving itself together to reveal the hidden truths of his own soul.

And so, Max embarked on a journey of self-discovery, guided by the mystical energies of the book and the wisdom of his own heart. As he explored the depths of his own inner world, he began to tap into a reservoir of creativity, courage, and compassion that he never knew he possessed. The boundaries between fantasy and reality blurred, and Max found himself walking the thin line between the two, where the magic of the book merged with the magic within him to create a world of limitless possibility.

**Embracing the Power Within**

As the days passed, Max grew more confident in his ability to harness the power of magic within himself. He practiced the spells and incantations, feeling the energy coursing through his veins like liquid fire. The world around him transformed, taking on a vibrant hue that reflected the colors of his own imagination. Friends and family marveled at the change in him, noticing the way he moved with a newfound sense of purpose and confidence.

With every step forward, Max felt himself becoming one with the universe, connected to the threads of fate that wove together the tapestry of existence. He understood that the true magic lay not in the book or its words, but in the depths of his own heart, where the spark of creation burned bright and true. And as he looked back on the day he discovered the magical book, Max smiled, knowing that he had unlocked a secret that would stay with him forever – the secret of embracing the power within himself, and unleashing a world of magic upon the world.

---

Next, we utilize the first two-thirds of the extended first half for ICL to extend the initial response. We use gray color here to indicate the ICL segment.

---

**Extended response (1071 words)**

**The Boy Who Unlocked His Inner Magic**

**A World of Discovery**

Twelve-year-old Max had always been captivated by the dusty, cobweb-covered trunks and boxes that filled his family's attic. For years, he had spent countless hours exploring the hidden treasures he found there, from vintage toys and antique clothes to mysterious artifacts and forgotten heirlooms. Each new discovery sparked a sense of wonder and curiosity within him, fueling his imagination and inspiring his love for adventure. His parents often joked that he had a knack for uncovering the most unlikely treasures, and they would smile knowingly whenever he emerged from the attic with some newfound gem.

But nothing could have prepared Max for what he was about to discover on that fateful Saturday morning. As he climbed the creaky stairs leading up to the attic, the warm sunlight streaming through the windows illuminated the space, casting a golden glow over the familiar landscape of trunks, boxes, and forgotten relics. With a spring in his step, Max set out to explore the attic once again, determined to uncover a treasure that would surpass all others.

**The Mysterious Book**

While rummaging through an old trunk, Max's fingers brushed against a peculiar-looking book bound in worn leather. The cover was adorned with intricate symbols and patterns that seemed to shimmer in the dim light of the attic, like a constellation of tiny stars. As he opened the book, a faint hum filled the air, and the pages crackled with an otherworldly energy. The musty scent of aged paper wafted up, carrying with it whispers of secrets and mysteries long forgotten. Intrigued, Max carefully turned the pages, revealing illustrations of fantastical creatures, magical spells, and ancient incantations that seemed to leap off the page and dance before his eyes.

A note on the first page caught his eye, written in elegant script that seemed to shimmer like moonlight:

"For those who seek the spark within, To unlock the secrets of the heart, Read these words, and listen well, For the magic lies within you, not in this spell."

As Max read the words, he felt an unusual tingling sensation in his fingers, as if the book was awakening something deep within him. It was as if the very essence of the book was resonating with his own heartbeat, calling forth a part of him that he never knew existed. Suddenly, visions began to flood his mind – vivid images of enchanted forests, shimmering stars, and majestic dragons soaring through the skies. The world around him melted away, leaving only the thrumming energy of the book and the promise of untold wonders waiting to be discovered.

**Unlocking the Secrets Within**

As Max delved deeper into the book, he began to realize that the words were not just mere ink on paper, but a key to unlocking the secrets of his own heart. The illustrations seemed to come alive, swirling with colors and textures that danced across the page like a kaleidoscope of dreams. The magical spells and incantations whispered secrets in his ear, guiding him toward a path of self-discovery and empowerment. With each passing moment, Max felt his connection to the book growing stronger, as if the very fabric of reality was weaving itself together to reveal the hidden truths of his own potential.

Over the next few days, Max returned to the attic again and again, devouring the knowledge contained within the book. He practiced the simple spells and incantations, feeling the power coursing through his veins like electricity. At first, it was just a gentle buzzing, but soon the sensations grew stronger, and Max started to notice changes within himself. Objects moved with a mere thought, levitated small items, and tiny sparks of light flickered to life in his fingertips.

His friends at school took notice of the transformation in him, too. They saw him effortlessly solving complex math problems or creating intricate drawings that seemed to come alive on the page. Some whispered among themselves about the "magic" that had taken hold of their classmate, but Max knew better. He understood that the true source of his newfound abilities lay not in the book, but within himself. The book had merely unlocked a door to his own inner potential, allowing him to tap into a reservoir of untapped magic that had been slumbering inside him all along.

**Mastering the Art of Magic**

As Max continued to hone his skills, he learned to control his powers more effectively, mastering techniques to heal injuries, communicate with animals, and even bend time and space to his will. His confidence soared, and he knew that he was no longer just an ordinary boy; he was a wizard, capable of achieving incredible feats. With each passing day, Max became more adept at wielding his powers, channeling them toward the greater good. He used his abilities to help those in need, whether it was calming a frightened animal or shielding his family from harm.

But with great power comes great responsibility, and Max understood that he must use his gifts wisely. He recognized that the line between good and evil was ever-shifting, and that even the most well-intentioned actions could have unintended consequences. With the weight of this understanding, Max vowed to walk a path of compassion and wisdom, using his powers to uplift and protect those around him.

**A Beacon of Hope**

As Max looked back on the journey that had brought him to this point, he knew that the true magic lay not in the pages of a book, but in the depths of his own heart and soul. The experience had taught him that the greatest magic of all was not the ability to cast spells or wield elemental forces, but the capacity to connect with others, to understand their struggles and triumphs, and to offer a helping hand when needed.

And so, with the magical book safely tucked away, Max embarked on a journey to explore the limits of his abilities, to learn from his mistakes, and to grow into the person he was meant to be. He knew that the road ahead would be fraught with challenges and uncertainties, but he also knew that he was ready. For Max had discovered a truth that would stay with him forever: the magic within him was not a gift, but a reflection of the infinite potential that lay within every individual, waiting to be unleashed.

After completing one inner iteration, we successfully extended the initial response from 553 to 1071 words. We then replaced the initial response with this extended version and conducted two more iterations. As a result, the response grew to 1325 words in the second iteration and reached 1538 words in the final extension.

**Final extended response (1538 words)**

**The Boy Who Unlocked His Inner Magic**
**A World of Discovery**

In the quaint town of Willowdale, where sun-kissed streets and charming homes blended seamlessly into the lush countryside, twelve-year-old Max had grown up surrounded by the nostalgic charm of his family's ancestral home. Nestled at the top of a gentle hill, the house boasted an expansive attic that seemed to stretch on forever, its wooden beams groaning under the weight of generations' worth of memories. For Max, the attic was a realm of endless fascination, a treasure trove of dusty relics and forgotten heirlooms waiting to be unearthed. Over the years, he had developed an uncanny ability to sniff out the most unlikely treasures, earning him the affectionate nickname "Treasure Hunter" among his friends and family.

His parents often marveled at his innate talent for discovery, watching with pride as he delved deeper into the attic's labyrinthine corridors, unearthing everything from antique dolls and vintage clothing to rare coins and mysterious artifacts. The thrill of the hunt was intoxicating, and Max reveled in the rush of adrenaline that accompanied each new find. Yet, despite the countless hours he'd spent exploring the attic, Max couldn't shake the feeling that there remained one last secret waiting to be uncovered – a hidden treasure that would surpass all others and reveal a piece of the family's history he'd never known existed.

This sense of longing had become an integral part of Max's identity, driving him to explore every nook and cranny of the attic with a fervor that bordered on obsession. He had scoured the shelves, rummaged through trunks, and even crawled through narrow crawlspaces in search of the elusive treasure. And yet, despite his unwavering dedication, the attic continued to yield its secrets slowly, teasing him with tantalizing glimpses of what might lie just beyond his grasp.

That fateful Saturday morning dawned bright and clear, with sunlight spilling through the attic windows like liquid gold. The air was alive with the sweet scent of blooming flowers and the distant chirping of birds, beckoning Max to begin his latest expedition. He climbed the creaky stairs, his footsteps echoing through the silence, and stepped into the attic with a sense of anticipation that bordered on excitement.

**The Attic's Hidden Depths**

As he ventured deeper into the attic, Max found himself navigating a maze of narrow pathways and dimly lit alcoves. Cobwebs clung to the corners, casting eerie shadows on the walls, while the faint scent of decay wafted through the air, hinting at the presence of long-forgotten relics. Despite the eerie atmosphere, Max felt an overwhelming sense of belonging, as if the attic had been waiting patiently for him to uncover its secrets.

He pushed aside a tattered curtain, revealing a hidden compartment filled with dusty vases, tarnished silverware, and faded photographs. A nearby trunk, adorned with intricate carvings, seemed to whisper secrets in the flickering light, drawing Max closer with an otherworldly allure. The trunk's lid creaked open with a soft sigh, releasing a puff of dust that carried the whispers of the past.

**The Mysterious Book**

Amidst the layers of yellowed tissue paper and faded lace, a small, leather-bound book lay nestled, its cover exuding an aura of mystique. The book itself was a work of art, crafted from supple leather that seemed to glow with an inner light. Delicate patterns etched into the surface shimmered like constellations on a clear night sky, drawing Max in with an otherworldly allure. As he lifted the cover, the pages rustled softly, releasing a fragrance that was both earthy and ethereal – the scent of aged parchment infused with hints of vanilla and sandalwood.

The musty aroma transported Max to a realm of ancient knowledge, where forgotten wisdom waited to be rediscovered. The pages themselves were a marvel, illustrated with fantastical creatures that seemed to spring to life beneath his fingertips. Dragons soared through the skies, their scales glinting like diamonds; unicorns pranced across moonlit meadows, their manes flowing like silk; and phoenixes burst forth from fiery ashes, reborn in a blaze of glory. Each illustration was a window into a world beyond the mundane, a realm where magic was woven into the very fabric of existence.

A note on the first page, penned in calligraphy that shone like moonlight, spoke directly to Max's heart:

> "For those who seek the spark within, To unlock the secrets of the heart, Read these words, and listen well,
> For the magic lies within you, not in this spell."

As Max read the words, a strange tingling sensation coursed through his fingers, as if the book was awakening a part of him that had lain dormant for centuries. The room around him dissolved, replaced by visions of enchanted realms and mystical landscapes that unfolded like a tapestry before his mind's eye. The thrumming energy of the book synchronized with his heartbeat, drawing him deeper into the mystery.

**Unlocking the Secrets Within**

With each passing moment, Max became increasingly enthralled by the book's secrets, pouring over the pages as if devouring a feast. The words on the pages began to take on a life of their own, speaking directly to his soul in a language that transcended words. The illustrations pulsed with color and texture, transforming into living, breathing entities that swirled around him like a maelstrom of creativity.

As he delved deeper into the book's mysteries, Max realized that the words were not mere ink on paper, but a key to unlocking the deepest recesses of his own heart. The book was teaching him that the true source of magic lay not in external sources, but within himself – a reservoir of untapped potential waiting to be tapped. With each passing day, Max felt the boundaries between reality and fantasy blurring, as if the very fabric of existence was being rewritten to accommodate his growth.

Over the course of several days, Max returned to the attic again and again, lost in the world of the book. He practiced the simple spells and incantations, feeling the power coursing through his veins like electric currents. At first, it was a gentle buzzing, but soon the sensations intensified, and Max noticed subtle yet profound changes within himself. Objects moved with a mere thought, levitating above the floorboards like delicate balloons; small sparks of light flickered to life in his fingertips, illuminating the darkness; and the very air around him seemed to vibrate with an otherworldly energy.

**The Awakening of Potential**

As Max harnessed the power of the book, he began to realize that he was not alone in his newfound abilities. His classmates at school took notice of the transformation, sensing a change in the way he approached challenges and interacted with the world around him. Some whispered about the "magic" that had taken hold of their classmate, while others looked on with a mixture of awe and trepidation.

Max, however, understood that the true nature of his powers lay far beyond the realm of magic tricks and illusions. He comprehended that the energies coursing through his body were a manifestation of his own potential, a testament to the boundless possibilities that lay within every individual. As he navigated the complexities of adolescence, Max learned to wield his abilities with care and precision, channeling them toward the greater good.

**Mastering the Art of Magic**

As Max honed his craft, he mastered techniques to heal wounds, communicate with animals, and even manipulate time and space to his will. His confidence soared, and he realized that he was no longer an ordinary boy; he was a wizard, capable of achieving wondrous feats. With each passing day, Max became more skilled at wielding his powers, directing them toward the greater good. He shielded his loved ones from harm, calmed frightened animals, and brought solace to those in need.

Yet, with great power comes great responsibility, and Max understood that the line between good and evil was ever-shifting, like the tides of a restless sea. Even the most benevolent intentions could lead to unforeseen consequences, and Max vowed to tread a path of compassion and wisdom, using his gifts to uplift and protect those around him.

**A Beacon of Hope**

Looking back on his journey, Max came to comprehend that the true magic lay not in the pages of a book, nor in the powers he wielded, but in the depths of his own heart and soul. The experience had taught him that the greatest magic of all was not the ability to conjure spells or command elemental forces, but the capacity to connect with others, to empathize with their struggles and triumphs, and to extend a helping hand when needed.

With the magical book safely stored away, Max embarked on a journey to explore the limits of his abilities, to learn from his mistakes, and to grow into the person he was destined to become. He knew that the path ahead would be strewn with challenges and uncertainties, but he also knew that he was prepared. For Max had discovered a truth that would remain with him forever: the magic within him was not a gift, but a reflection of the boundless potential that resides within every individual, waiting to be unleashed.

