# OpenReview forum: "Language Models can Self-Lengthen to Generate Long Texts"
_ICLR.cc/2025/Conference — ICLR 2025 Conference Withdrawn Submission_

### Official Review · Reviewer_srD7 · 2024-10-17

**Soundness:** 2
**Presentation:** 3
**Contribution:** 1
**Rating:** 3
**Confidence:** 5

**Summary:**

This paper introduces Self-Lengthen, an iterative training framework designed to enhance the long-text generation capabilities of large language models (LLMs) without auxiliary data or proprietary models. The framework employs a Generator and an Extender to iteratively extend model outputs, resulting in progressively longer and more refined responses. The experimental results show that Self-Lengthen outperforms existing methods like instruction backtranslation and behavior imitation, particularly in long-text generation tasks. The paper also provides a thorough empirical evaluation, including human assessments and benchmark comparisons.

**Strengths:**

1, The proposed Self-Lengthen framework introduces a unique iterative approach to improve long-text generation by utilizing the intrinsic capabilities of LLMs without relying on additional external datasets or proprietary models.

2, The method is simple yet practical, focusing on leveraging existing models' capabilities through iterative extension. This makes the method easy to implement and potentially scalable to various domains where long-text generation is required.

3, The authors conduct extensive evaluations using different open-source models and comparison baselines. The experiments are well-designed to demonstrate the improvements in generating longer outputs while maintaining acceptable quality.

**Weaknesses:**

1. **Motivation**: The motivation for this work is not sufficiently compelling. Regarding the instruction backtranslation method, SlimPajama already provides a large amount of long-text data generated by real-world [1]. For the behavior imitation approach, it is difficult to agree that there is a significant difference between using GPT-4 and open-source models. The LongWrite's agentwrite method can also use open-source models to generate data, which undermines the claimed uniqueness of Self-Lengthen.

2. **Methodology Design**: The Self-Lengthen method, while simple, may not be entirely reasonable. When humans perform long-text tasks, they typically do not opt for multiple iterative expansions. The multiple extension approach seems different to agentwrite from LongWrite, where an outline is first provided, and then expanded. It is unclear why the former approach would yield better performance compared to the latter. Providing a justification or empirical evidence to support this claim would strengthen the argument.

3. **Evaluation Metrics**: For long-text tasks, people often prioritize the quality of generated text over whether the length requirements are strictly met. Regarding the Output Quality Score metric, I have concerns about relying on LLMs for quality evaluation.
   - **Subjectivity**: Much of creative writing inherently involves strong subjective elements, especially concerning creativity. Using LLMs to evaluate such content may not capture the nuances of human judgment.
   - **Limitations of LLMs**: While LLMs are effective at long-text retrieval tasks, I believe they are still insufficient for fully understanding the content of long documents, particularly regarding holistic comprehension. Relying entirely on LLMs for evaluation may not be appropriate.



[1] [SlimPajama: A 627B Token Cleaned and Deduplicated Version of RedPajama](https://cerebras.ai/blog/slimpajama-a-627b-token-cleaned-and-deduplicated-version-of-redpajama)
[2] *LongWriter: Unleashing 10,000+ Word Generation from Long Context LLMs*

**Questions:**

1. **Comparison with LongWrite**: The differences between Self-Lengthen and LongWrite are not clearly articulated. For instance, both methods use metrics like length-following and quality. The paper should provide a more explicit comparison to distinguish itself from LongWrite, including discussing any unique advantages or limitations.
2.  Figure 6: The figure appears to have display issues and should be corrected for better readability.

---

### Official Review · Reviewer_pQCD · 2024-10-21

**Soundness:** 2
**Presentation:** 1
**Contribution:** 2
**Rating:** 3
**Confidence:** 4

**Summary:**

This paper proposes an iterative training framework called Self-Lengthen to enable LLMs to generate long responses, without the need for auxiliary data or proprietary models. The framework consists of two roles: the Generator produces the initial response, which is then split and expanded by the Extender to construct a new training set with longer responses. This process can be done iteratively to make the model handle increasingly longer responses.

**Strengths:**

1. Compared to the previous method, Self-Lengthen has no need for auxiliary data or powerful proprietary models, and supports outputs with more diverse styles and types.
2. Experiments on benchmarks and human evaluations show that Self-Lengthen outperforms existing methods in long-text generation when applied to top open-source LLMs such as Qwen2 and LLaMA3.

**Weaknesses:**

1. The design of LonGen benchmark is too similar to the benchmark in LongWriter (i.e.,  LongBench-Write), and many tables (e.g., Table 2, 3) and figures (e.g., Fig 5, 6) are similar to those in LongWriter without proper citations. The authors should give a more detailed explanation and comparison.

2. There are many missing details in the experiments, including the calculation method of distinct scores, the training data statistics, and the supported maximum output length.

3. The length control ability (left of Fig. 1, not well aligned with the desired length)  and maximum reported output length (8k) of the trained models are relatively poor compared to the SoTA methods such as LongWriter.

**Questions:**

1. How do you split the response $y$ into $y[:1/2]$ or $y[:2/3]$? Is this based on the number of tokens or what other criteria?
2. How are the distinct scores in human evaluation calculated?
3. In human evaluation, why select different queries for P&W and Slef-Lengthen? Comparing responses to the same queries seems more reasonable.
4. What are the source and statistics of the training data? The paper only mentions they utilize user logs without further details.
5. What is the total number of iterations of the trained models?
6. Why does Section 5.2 say that "LongWriter datasets either lack appropriate length requirements in their queries or have unsuitable length constraints, resulting in results that fall significantly short of expectations."? As I know, LongWrite dataset has length constraints in the queries. Are there any evaluation results of directly using the LongWriter dataset or their open-sourced models such as LongWriter-llama3.1-8b to support the claim?

**Details Of Ethics Concerns:**

The design of LonGen benchmark is too similar to the benchmark in LongWriter (i.e.,  LongBench-Write), and many tables (e.g., Table 2, 3) and figures (e.g., Fig 5, 6) are similar to those in LongWriter without proper citations. The authors should give a more detailed explanation and comparison.

---

### Official Review · Reviewer_UANm · 2024-10-27

**Soundness:** 1
**Presentation:** 2
**Contribution:** 2
**Rating:** 3
**Confidence:** 4

**Summary:**

The paper proposes a new method Self-Lengthen to tackle long-form generation, which existing large language models (LLMs) often fall short. The proposed method involves iteratively training the Generator and the Extender on progressively long responses generated by the model from the previous iteration. To produce these long responses, prompting and rule-based methods are introduced to guarantee the output length and quality. Experimental results show that the proposed method can extend the response length of qwen2-7b and llama3.1-8b and outperform the existing methods such as instruction back-translation and plan-and-write.

**Strengths:**

- This paper proposes a new method to improve language models’ long-form generation performance.
- This topic is relevant to a wide range of applications which are bottlenecked by the response length that language models can reliably output.

**Weaknesses:**

- The proposed method contains a seemingly arbitrary decision of truncating the response to ½ or ⅔ for further extension. It is unclear why these cutoffs were chosen and how they compare to other cutoffs.
- The proposed method utilized surface form heuristics (e.g. length, repetition)  to ensure the quality of extended responses, while the semantic content is not quality assured. It is unclear if training on synthetic self-generated data hurts other LM capabilities, e.g., math/code reasoning and generating factual content.
- It is unclear if the proposed method hurts the short-form generation performance.
- In Section 5.1 “Due to the extraordinarily long length of the responses, we utilized GPT-4o to summarize and analyze them, aiding evaluators in their assessments for improved readability and increased annotation efficiency”. Introducing proprietary models into human evaluation is not preferable as the model-introduced error cannot be reliably detected and can further complicate human evaluation results.
- In section 5.2 “To ensure a fair evaluation, we introduce length constraints by applying the same workflow and prompts to their original queries. “ It is unclear if the proposed method is better because the baselines are corrupted by these additional steps.
- The total evaluation set is quite small (240 examples) and it is unclear how statistically significant the improvement is (e.g. 85.82 vs 85.52). I do notice llama3.1 with backtranslation performs significantly worse than the counter part of qwen. However, no extensive explanation is provided.
- Overall, I believe the paper can be improved by clarifying some experimental and evaluation details.

**Questions:**

- Why did you choose to truncate at  ½ and ⅔ in the micro iteration? Is there additional evidence showing this decision is empirically the best? If so, any conjecture why this is the case?
Does this method affect short-form generation capability?
- Can you provide more explanation about the distinct score? The responses of the backtranslation method are written by humans. Can you explain why its distinct score is lower than self-lengthen?
- Can you give more explanation for why qwen2 and llama3 performance differ significantly for the backtranslation baseline?
- Can you estimate how significant the gains of the proposed method is?

---

### Official Review · Reviewer_Z4GE · 2024-11-04

**Soundness:** 3
**Presentation:** 3
**Contribution:** 3
**Rating:** 6
**Confidence:** 4

**Summary:**

This paper proposes an innovative iterative training framework called Self-Lengthen, which consists of two roles: a generator and an extender. The generator produces an initial response, which is then split and extended by the extender. This process generates new, longer responses that are used to repeatedly train both the generator and the extender. As a result, the model is gradually trained to handle increasingly longer responses.

**Strengths:**

1. Self-Lengthen is cost-effective and easy to use. It only requires a set of seed instructions for long-text output tasks and an open-source instruction model to automatically enhance the model's ability to generate long-text outputs.

2. This paper proposes a two-stage extension method that ensures the extension does not end normally. This creates space for the model to seamlessly connect preceding and succeeding segments, thereby enhancing its ability to complete extension tasks.

3. The author provided the complete code, ensuring high reproducibility.

**Weaknesses:**

1. The paper adopts a rule-based approach to filter out invalid responses to ensure their quality. How is 'frequent repetition' specifically determined? Is it based on rules or scored by a more advanced LLM? If the extended responses merely describe the same meaning in different styles, is this kind of extension meaningful?

2. During the process of instruction evolution, some instruction data are inherently unsuitable for this type of evolution. For example, if the response to an instruction is a proper noun, extending it might generate some meaningless content. Is such data helpful for training? Or, even if it gets filtered out during the evolution process, it still wastes computational resources. So, when collecting seed data, can we pre-filter these unsuitable instructions based on certain methods to save costs?

3. Hope to get an analysis of the GPU hours required for the Self-Lengthen extension in order to discuss its efficiency issues.

**Questions:**

please see weaknesses

---

### Author Response · Authors · 2024-11-24
**Official Rebuttal by Authors**

We are grateful for the valuable feedback from all the reviewers. Although some comments appear to be of lower quality and somewhat unfair, some other insights have significantly helped us enhance the paper, taking it to a noticeably higher level in our latest version,  Additionally, we also address and clarify some misunderstandings mentioned in the reviews here.

- Comparison with LongWriter (Reviewer UANm, pQCD, srD7): First, LongWriter is a **CONTEMPORARY WORK** with us, released within a time period of two months, so we are not fully eligible to compare with them in all details. Second, we **offer a very clear comparison in the Prerequisite Section Table 1**: 1) We do **not require starting with a robust Long-Context LLM** (we achieve this iteratively), and 2) we can produce responses in a **more diverse form**. Additionally, while LongWriter uses a straightforward agent framework, our approach is superior in terms of **methodological novelty**.

    In addition, regarding the evaluation logic similar to LongWriter (Reviewer pQCD), it’s worth noting that a paper contributing a method should not have the eligibility to have a totally different evaluation approach compared to others. For consistent tasks, **evaluations remaining similar is normal**. Moreover, we noticed that the length-following categories in LongWriter are too limited, often just categorized as 'about.' To address this, we've **expanded the categories** to include 'range,' 'about,' 'above,' and 'below,' and we've developed specific scoring metrics for each type. This ensures a more comprehensive length-following evaluation that better aligns with real user needs. We will incorporate this discussion and clearly state our adherence to a similar evaluation logic with LongWriter.

- Explanation on 1/2 and 2/3: We use 1/2 to effectively double the length of the response with each iteration. Our results demonstrate that this approach aligns well with our expectations. The technique of dropping the last part has proven invaluable for seamlessly connecting two parts, and we have found that removing 1/3 is an optimal proportion for achieving this.

- Incorporating GPT-4o for summarizing and analyzing responses to aid human evaluators (Reviewer UANm): Scoring responses entirely by humans is **not practical**. Once you attempt to label an extremely long response, you'll realize the complexity of the task. By utilizing the third-party LLM GPT-4o for summarization and analysis, we can **substantially decrease human effort**. We clarify in the instructions that these summaries and analyses are **for reference only and should not influence their independent judgment.** (Appendix F.1). Additionally, we believe this evaluation method could be adopted in future research on assessing lengthy outputs.

- Adopting same length control for Suri, LongWriter, and ours (Reviewer pQCD): we demonstrate clearly in Section 5.2 that their responses fail to meet appropriate length requirements (Suri) in their queries or apply unsuitable length constraints (LongWriter). If you have doubts, you can verify this by examining their official dataset on HuggingFace. Our evaluation incorporates a much more diverse range of length-following types, rather than focusing solely on terms like 'about.' Additionally, we discover that **implementing the same workflow can considerably enhance their performance**. We **strive to ensure a fair judgment**, and it's unfortunate that some have tried to unjustly accuse us of having ulterior motives.

- Distinct Score Metric (Reviewer UANm, pQCD): The Distinct Score is a well-known metric in NLP used to assess the diversity of responses, introduced by the paper "A Diversity-Promoting Objective Function for Neural Conversation Models."

- Motivation and Methodology Design (Reviewer srD7): See comparisons between Suri and LongWriter.

- Limitations of LLMs in Evaluation (Reviewer srD7): Please check that we also have a human evaluation section.

- Figure 6 display issues (Reviewer srD7): No display issues.

- Lower length control ability (Reviewer pQCD): In the Experiment Section, we **clearly demonstrate that we achieve state-of-the-art performance in length control**. Our results are based on the entire evaluation set. In contrast, the LongWriter paper misleads readers by selectively **testing on a few chosen queries** (LongWrite-Ruler) to produce a seemingly well-aligned figure, which presents an inflated outcome that does not accurately reflect their true capability.

- Acquiring additional details (Reviewer pQCD, UANm, Z4GE): Some of these details have **already been presented** in our paper: We conducted a total of three macro-iterations, as discussed in Sections 5.2 and 6. Additionally, we utilized the initial LLM to validate the generated instructions, discarding any unsuitable ones for producing lengthy responses (Section 4.1). We will clarify these points more clearly and provide more detailed information in our latest version.

---

### Note · Authors · 2024-11-24

I have read and agree with the venue's withdrawal policy on behalf of myself and my co-authors.